# Constitutive XBP-1s-mediated activation of the endoplasmic reticulum unfolded protein response protects against pathological tau

Sarah M. Waldherr[1,2], Timothy J. Strovas [3], Taylor A. Vadset[2], Nicole F. Liachko[2,3] & Brian C. Kraemer [1,2,3,4,5]

To endure over the organismal lifespan, neurons utilize multiple strategies to achieve protein homeostasis (proteostasis). Some homeostatic mechanisms act in a subcellular compartment-specific manner, but others exhibit trans-compartmental mechanisms of proteostasis. To identify pathways protecting neurons from pathological tau protein, we employed a transgenic *Caenorhabditis elegans* model of human tauopathy exhibiting proteostatic disruption. We show normal functioning of the endoplasmic reticulum unfolded protein response (UPR[ER]) promotes clearance of pathological tau, and loss of the three UPR[ER] branches differentially affects tauopathy phenotypes. Loss of function of *xbp-1* and *atf-6* genes, the two main UPR[ER] transcription factors, exacerbates tau toxicity. Furthermore, constitutive activation of master transcription factor XBP-1 ameliorates tauopathy phenotypes. However, both ATF6 and PERK branches of the UPR[ER] participate in amelioration of tauopathy by constitutively active XBP-1, possibly through endoplasmic reticulum-associated protein degradation (ERAD). Understanding how the UPR[ER] modulates pathological tau accumulation will inform neurodegenerative disease mechanisms.

[1] Molecular and Cellular Biology Interdisciplinary Program, University of Washington, Seattle, WA 98195, USA. [2] Division of Gerontology and Geriatric Medicine, Department of Medicine, University of Washington, Seattle, WA 98104, USA. [3] Geriatrics Research Education and Clinical Center, Veterans Affairs Puget Sound Health Care System, Seattle, WA 98108, USA. [4] Department of Psychiatry and Behavioral Sciences, University of Washington, Seattle, WA 98195, USA. [5] Department of Pathology, University of Washington, Seattle, WA 98195, USA. Correspondence and requests for materials should be addressed to B.C.K. (email: kraemerb@u.washington.edu)

Persisting for decades, human neurons face unique metabolic challenges, including the need to maintain long-term protein quality control. Cellular systems such as the ubiquitin-proteasome system (UPS), autophagy, the heat shock response, and the unfolded protein responses in the endoplasmic reticulum (UPR$^{ER}$) and mitochondria (UPR$^{mt}$) enable neurons to maintain protein homeostasis (proteostasis) within different cellular compartments. Proper folding of all secretory proteins occurs within the endoplasmic reticulum (ER), comprising ~25% of all proteins in the mammalian proteome[1,2]. Proteostasis imbalance in the ER triggers the UPR$^{ER}$, which causes the molecular chaperone binding immunoglobulin protein (BiP) to dissociate from the three ER transmembrane stress sensors [protein kinase RNA-like ER kinase (PERK), inositol-requiring enzyme 1 α (IRE1α), and activating transcription factor 6 (ATF6)] (Fig. 1a). This results in immediate inhibition of general translation to decrease the protein folding load in the ER and activation of genes for restoring protein folding homeostasis. PERK attenuates global translation to reduce entrance of newly synthesized proteins into the ER lumen and activates translation of activating transcription factor 4 (ATF4), which controls the expression of genes involved in autophagy, amino acid metabolism, antioxidant responses, and apoptosis[3]. IRE1α catalyzes non-canonical splicing of X-box binding protein 1 (XBP1) mRNA into the constitutively active form XBP1s, which becomes the master UPR$^{ER}$ transcription factor controlling a wide range of gene targets required for ER proteostasis. These include genes related to protein folding, ER-associated protein degradation (ERAD), protein translocation into the ER, and lipid synthesis[4]. ATF6 translocates to the Golgi apparatus where it is cleaved into C- and N-terminal fragments. The N-terminal fragment (ATF6n) translocates to the nucleus to activate transcription of ERAD genes and UPR$^{ER}$ master transcription factor XBP1s[3]. If the stress is prolonged and unresolved, the UPR$^{ER}$ switches from a pro-survival role to a pro-apoptotic role, activating downstream signaling events of all three stress sensors to induce cell death.

The UPR$^{ER}$ is an evolutionarily conserved adaptive signaling pathway. The most conserved UPR$^{ER}$ signaling branch, and the only one present in yeast, is initiated by IRE1α[5,6]. After BiP dissociates from the ER luminal domain of IRE1α, IRE1α undergoes dimerization and trans-autophosphorylation to activate its cytosolic endoribonuclease activity. Activated IRE1α catalyzes the excision of a short non-canonical intron from XBP1 mRNA. This causes a frameshift in the XBP1 coding sequence, translation into XBP1s protein, and entrance into the nucleus as an active transcription factor. UPR$^{ER}$ malfunctions are associated with a wide range of disease states, including tumor progression and diabetes, as well as immune, inflammatory, and neurodegenerative diseases[7]. Although the UPR$^{ER}$ plays many important physiological roles, we still do not fully understand the role of the UPR$^{ER}$ in post-mitotic neuronal survival.

Aging leads to a decline in proteostasis capacity, which can particularly affect post-mitotic cells such as neurons. Aging-related declines in proteostasis contribute to abnormal protein aggregation and downstream cellular degeneration. In neurodegenerative disease, age-dependent pathological protein accumulation drives synaptic impairment and neuronal loss. Several studies have highlighted the importance of the UPR$^{ER}$ in the aging brain. Under basal conditions, UPR$^{ER}$ chaperones such as BiP are decreased in several brain regions of aged mice[8–10]. When these animals experience cellular stress, the UPR$^{ER}$ fails to properly activate, leading to the upregulation of apoptotic pathway mediators[8–10]. The IRE1α/XBP1 branch may be particularly critical during aging. Dietary restriction extends lifespan in multiple organisms and requires IRE-1/XBP-1 activity in

Caenorhabditis elegans (C. elegans)[11]. Interestingly, constitutive XBP-1s activation in neurons can extend lifespan in C. elegans via XBP-1s-mediated non-cell autonomous signaling events[12]. Age strongly drives risk for many protein aggregate-deposition diseases of the nervous system; UPR$^{ER}$ dysfunction has been implicated in several of these diseases, including Alzheimer's, Parkinson's, and Huntington's disease, and amyotrophic lateral sclerosis[7,13].

Pathological tau protein serves as both a marker of brain aging and a hallmark of several neurodegenerative diseases termed tauopathies. Tauopathies exhibit cytoplasmic accumulation of neuronal and/or glial lesions containing hyperphosphorylated tau. As the most prevalent tauopathy, Alzheimer's disease (AD) neuropathology consists of both extracellular amyloid plaques composed of fibrillar amyloid beta (Aβ) peptides and intracellular neurofibrillary tangles composed of pathological tau protein. Other tauopathies include frontotemporal dementia with parkinsonism linked to chromosome 17 (FTDP-17), frontotemporal lobar degeneration (FTLD-tau), progressive supranuclear palsy (PSP), corticobasal degeneration (CBD), and Pick's disease[14]. Mutations in the microtubule-associated protein tau (MAPT) gene encoding the protein tau cause FTDP-17, providing a direct link between tau dysfunction and disease[15–17]. Interventions for tauopathies are limited to treatment of symptoms, as no approved therapies impact tau pathology or concomitant neurodegeneration.

To study tauopathies in a genetically tractable system, we have generated transgenic C. elegans that express human tau pan-neuronally[18]. These models exhibit behavioral abnormalities, accumulation of insoluble phosphorylated tau protein, progressive neurodegeneration, and shortened lifespan. We previously conducted a genome-wide RNA interference (RNAi) screen to identify modifiers of tau toxicity[19]. From this screen, we found RNAi targeting the UPR$^{ER}$ master transcription factor xbp-1 enhanced FTDP-17 mutant tau-induced behavioral deficits, and strong xbp-1 loss of function causes synthetic lethality with tau expression, nominating the UPR$^{ER}$ as a regulator of tau proteostasis[19]. The relationship between UPR$^{ER}$ function and pathological tau accumulation remains poorly characterized. To study the consequences of UPR$^{ER}$ loss of function on pathological tau, we have utilized loss of function mutations and gain of function transgenes to genetically dissect the role of the UPR$^{ER}$ in tau proteostasis.

## Results

**UPR$^{ER}$ branches differentially affect tauopathy phenotypes.** The UPR$^{ER}$ consists of three branches that regulate signaling to the nucleus and control the responses to unfolded protein within the ER lumen (Fig. 1a). Using the genetically tractable model organism C. elegans, we examined the effects of loss of function mutations ablating signaling through each of the three branches of the UPR$^{ER}$ [pek-1 (−/−), atf-6 (−/−), and xbp-1 (−/−)] on tau-mediated behavioral dysfunction in a mild model of human wildtype tau toxicity, which will be referred to as Tau (low) transgenic C. elegans[20]. Compared to non-transgenic (non-Tg) animals, Tau (low) animals exhibit mild behavioral deficits and no significant accumulation of pathological tau species.

C. elegans with a pek-1 loss of function mutation [pek-1 (−/−)] had normal locomotion (Supplementary Fig. 1a). If pek-1 regulates tau proteostasis, we would expect Tau (low); pek-1 (−/−) animals to exhibit an enhancement of tau-dependent behavioral dysfunction. However, when crossed with Tau (low) animals, pek-1 (−/−) did not worsen Tau (low) behavioral defects (Fig. 1b), suggesting the PEK-1 branch of the UPR$^{ER}$ does not mediate tauopathy phenotypes in C. elegans.

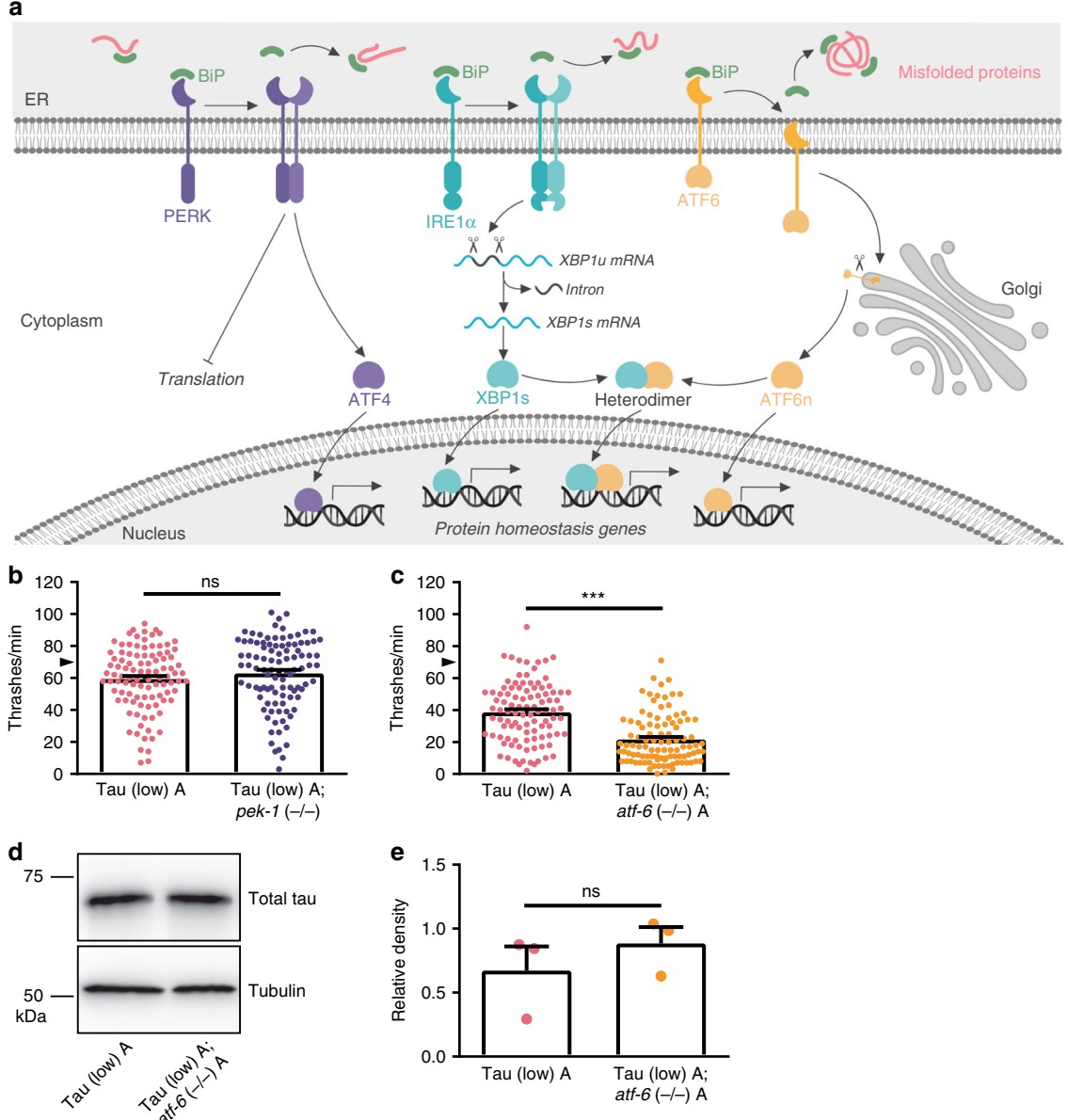

**Fig. 1** The UPR[ER] PEK-1/PERK and ATF-6/ATF6 branches are differentially involved in tauopathy in transgenic *C. elegans*. **a** Model figure of the mammalian UPR[ER], which acts to restore ER proteostasis. Cellular stress can disrupt ER proteostasis, triggering the UPR[ER] stress response. Three transmembrane stress sensors (PERK, IRE1α, and ATF6) are bound by the ER molecular chaperone BiP, holding them in inactive conformations. Misfolded proteins in the ER causes dissociation of BiP from the stress sensors, allowing adoption of active conformations. PERK dimerization and autophosphorylation allows phosphorylation of eIF2α and other substrates in the cytoplasm. Active PERK attenuates overall protein translation to reduce nascent protein folding load in the ER and activates transcription factor ATF4 to control target genes involved in restoring proteostasis. IRE1α dimerization promotes non-canonical splicing of *XBP1* mRNA in the cytoplasm to produce *XBP1s* mRNA encoding the UPR[ER] master transcription factor XBP1s. ATF6 translocates to the Golgi apparatus for cleavage into N- and C-terminal fragments, with the N-terminal fragment becoming an active transcription factor (ATF6n). Additionally, XBP1s and ATF6n can form heterodimers to regulate transcriptional target genes essential for ER proteostasis restoration. **b** *pek-1* loss of function in Tau (low) animals does not alter mild behavioral defects observed in a liquid environment [$n = 100$ animals; $N = 5$ biologically independent experiments; statistical analysis is by unpaired *t*-test, two-tailed (ns: $p = 0.2261$)]. **c** *atf-6* loss of function in Tau (low) animals enhances mild behavioral defects observed in a liquid environment [$n = 100$ animals; $N = 5$ biologically independent experiments; statistical analysis is by unpaired *t*-test, two-tailed (***$p < 0.0001$)]. **d**, **e** *atf-6* loss of function in Tau (low) animals does not affect soluble tau protein levels. Representative immunoblots for total tau and tubulin are shown, and densitometry analysis of chemiluminescence signals for tau normalized to tubulin are plotted [$N = 3$ biologically independent experiments; statistical analysis is by paired *t*-test, two-tailed (ns: $p = 0.4041$)]. All bar graphs represent mean + SEM. Arrowhead on *y*-axis of all liquid thrashing bar graphs denotes non-Tg animals average ~70 thrashes/min under standard laboratory conditions. Source data are available as a Source Data file

*C. elegans* with an *atf-6* loss of function mutation [*atf-6* (−/−)] had normal locomotion behavior (Supplementary Fig. 1b). However, Tau (low); *atf-6* (−/−) animals displayed worsened behavioral impairment compared to Tau (low) *C. elegans* (Fig. 1c).

To investigate whether *atf-6* loss of function in a Tau (low) background altered tau levels, we analyzed total tau protein expression by immunoblot. We found *atf-6* loss of function did not significantly affect total tau protein levels (Fig. 1d, e).

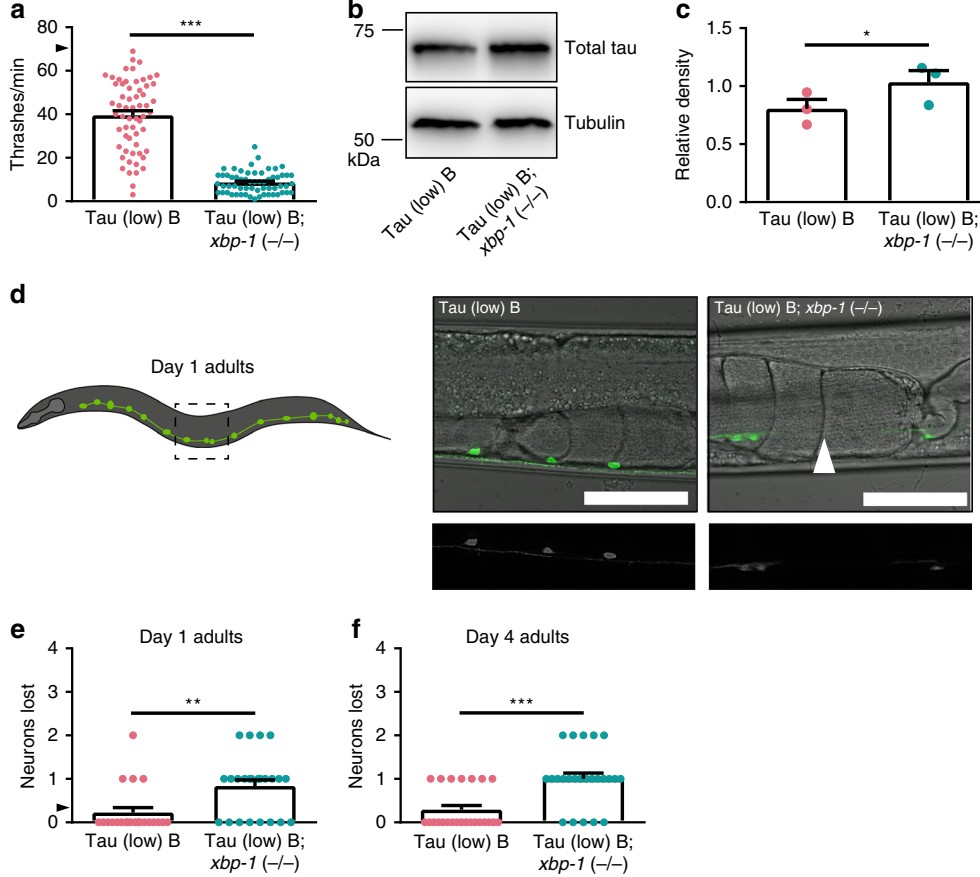

**Fig. 2** The UPR$^{ER}$ XBP-1 branch loss of function enhances tauopathy in transgenic *C. elegans*. **a** *xbp-1* loss of function in Tau (low) animals enhances mild behavioral defects observed in a liquid environment [$n = 60$ animals; $N = 4$ biologically independent experiments; statistical analysis is by unpaired *t*-test, two-tailed (***$p < 0.0001$)]. **b, c** *xbp-1* loss of function in Tau (low) animals increases soluble tau protein levels. Representative immunoblots for total tau and tubulin are shown, and densitometry analysis of chemiluminescence signals for tau normalized to tubulin are plotted [$N = 3$ biologically independent experiments; statistical analysis is by paired *t*-test, two-tailed (*$p = 0.0314$)]. **d–f** *xbp-1* loss of function in Tau (low) animals increases neuronal loss with age. **d** Representative images of D-type GABAergic ventral nerve cord neurons at day one of adulthood. The typical Tau (low) animal did not exhibit neuronal loss, while the Tau (low); *xbp-1* (−/−) animal exhibited loss of one neuron (white arrowhead = neuron loss; scale bar = 50 μm). **e, f** The number of D-type GABAergic ventral nerve cord neurons lost at day one of adulthood [$n = 22$ and 24 animals, respectively; $N = 2$ biologically independent experiments; statistical analysis is by unpaired *t*-test, two-tailed (**$p = 0.0020$)] and day four of adulthood [$n = 24$ animals; $N = 2$ biologically independent experiments; statistical analysis is by unpaired *t*-test, two-tailed (***$p < 0.0001$)] are plotted. All bar graphs represent mean + SEM. Arrowhead on *y*-axis of all liquid thrashing bar graphs denotes non-Tg animals average ~70 thrashes/min under standard laboratory conditions. Source data are available as a Source Data file

Therefore, *atf-6* loss of function enhances tau locomotion dysfunction independent from a role in tau protein turnover.

*C. elegans* with an *xbp-1* loss of function mutation [*xbp-1* (−/−)] had mild locomotion defects when compared to non-Tg animals (Supplementary Fig. 2); however, Tau (low); *xbp-1* (−/−) *C. elegans* exhibited dramatically exacerbated tau-dependent locomotion dysfunction (Supplementary Fig. 3). The mild behavioral deficits seen in Tau (low) animals were significantly enhanced with *xbp-1* loss of function. This phenotype was confirmed using an independent tau transgenic *C. elegans* (Fig. 2a). To determine the effects of *xbp-1* loss of function on tau protein turnover, we assayed tau protein levels by immunoblot. Tau (low); *xbp-1* (−/−) animals exhibited increased tau protein compared to Tau (low) animals (Fig. 2b, c).

Neurodegeneration is the ultimate consequence of pathological tau in tauopathy disorders. To assess whether *xbp-1* loss of function promotes tau-dependent neurodegeneration, we used a transgenic *C. elegans* reporter strain expressing green fluorescent protein (GFP) in D-type gamma-aminobutyric acid (GABA) positive neurons (*unc-47p::gfp*)[21]. This strain allows in vivo assessment of the integrity of GABAergic motor neurons along

the ventral nerve cord. At day 1 of adulthood, Tau (low) animals did not exhibit significant neurodegeneration (Fig. 2d, e). Fewer than one in four Tau (low) animals were missing one neuron (Fig. 2e). However, at day 1 of adulthood, *xbp-1* (−/−) exacerbated neurodegeneration in Tau (low) animals, including loss of neuronal cell bodies and breaks in the ventral nerve cord (Fig. 2d). Nearly every Tau (low); *xbp-1* (−/−) animal exhibited neurodegeneration (Fig. 2e). Loss of GABAergic motor neurons was more pronounced at day 4 of adulthood (Fig. 2f). Importantly, *xbp-1* (−/−) animals did not exhibit neurodegeneration on their own in the absence of tau (Supplementary Fig. 4). Taken together, *xbp-1* loss of function exacerbates tau-mediated behavioral deficits, increases accumulation of tau protein, and drives tau-dependent neurodegeneration in *C. elegans* models of tauopathy.

**Neuronal *xbp-1s* activation ameliorates tauopathy phenotypes.** Given *xbp-1* loss of function enhances tauopathy phenotypes, we hypothesized *xbp-1* gain of function might suppress tauopathy. To address this question, we utilized a severe model of human

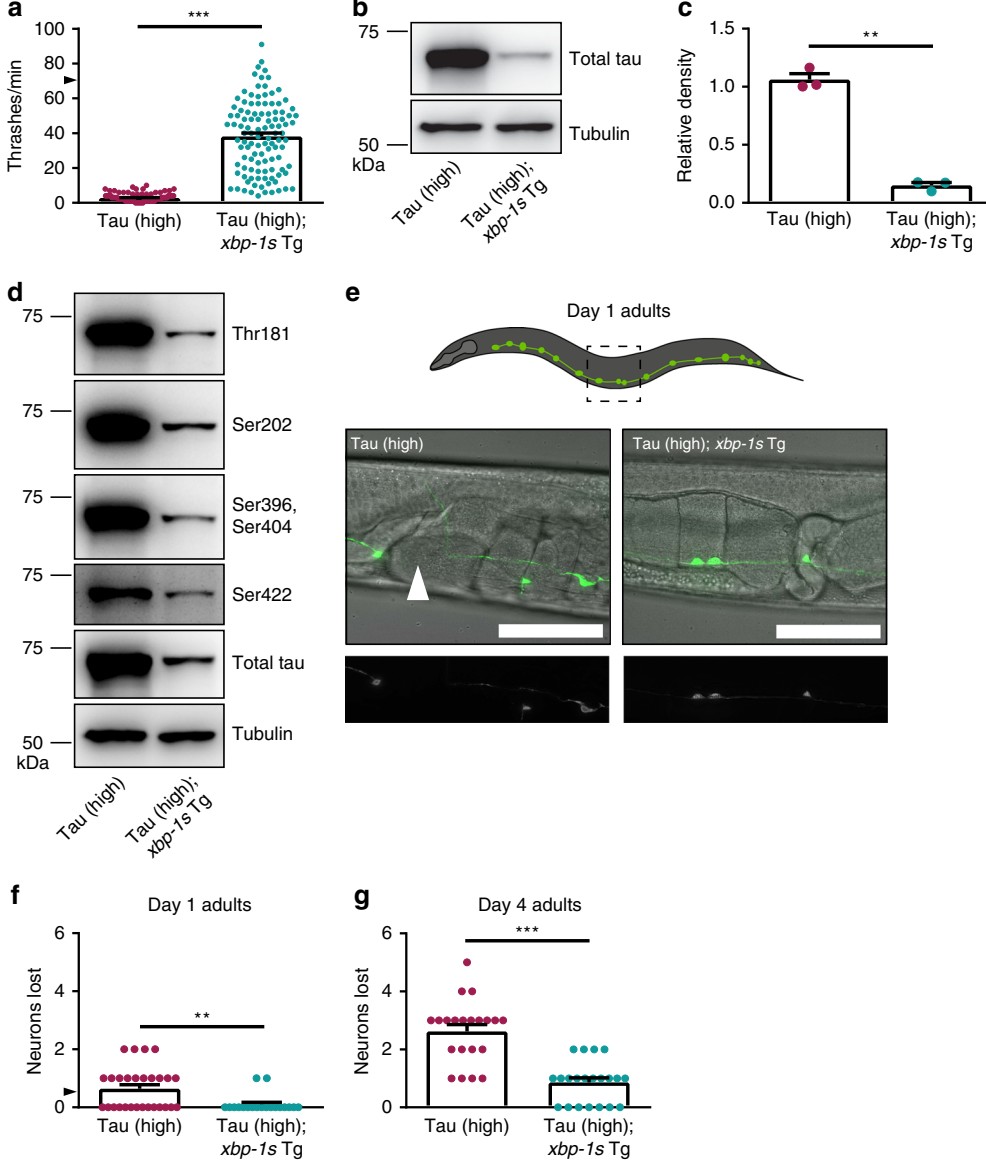

**Fig. 3** Constitutive UPR^ER activation by neuronal overexpression of *xbp-1s* suppresses tauopathy in transgenic *C. elegans*. **a** Neuronal overexpression of *xbp-1s* in Tau (high) animals suppresses severe behavioral defects observed in a liquid environment [$n = 110$ animals; $N = 6$ biologically independent experiments; statistical analysis is by unpaired *t*-test, two-tailed (***$p < 0.0001$)]. **b–d** Neuronal overexpression of *xbp-1s* in Tau (high) animals decreases different soluble tau protein species. **b**, **c** Representative immunoblots for total tau and tubulin are shown, and densitometry analysis of chemiluminescence signals for total tau normalized to tubulin are plotted [$N = 3$ biologically independent experiments; statistical analysis is by paired *t*-test, two-tailed (**$p = 0.0022$)]. **d** Representative immunoblots for phosphorylated tau (Threonine 181, Serine 202, Serine 396/Serine 404, Serine 422 phosphorylation sites), total tau, and tubulin are shown. Densitometry analysis of chemiluminescence signals for phosphorylated tau species normalized to total tau and tubulin are provided in Supplementary Table 3 ($N = 4$ biologically independent experiments; statistical analysis is by paired *t*-test, two-tailed). **e–g** Neuronal overexpression of *xbp-1s* in Tau (high) animals decreases neuronal loss with age. **e** Representative images of D-type GABAergic ventral nerve cord neurons at day one of adulthood. The typical Tau (high) animal exhibited loss of one neuron, while the Tau (high); *xbp-1s* (Tg) animal did not exhibit neuronal loss (white arrowhead = neuron loss; scale bar = 50 µm). **f**, **g** The number of D-type GABAergic ventral nerve cord neurons lost at day one of adulthood [$n = 28$ and 19 animals, respectively; $N = 2$ biologically independent experiments; statistical analysis is by unpaired *t*-test, two-tailed (**$p = 0.0042$)] and day four of adulthood [$n = 21$ animals; $N = 2$ biologically independent experiments; statistical analysis is by unpaired *t*-test, two-tailed (***$p < 0.0001$)] are plotted. All bar graphs represent mean + SEM. Arrowhead on *y*-axis of all liquid thrashing bar graphs denotes non-Tg animals average ~70 thrashes/min under standard laboratory conditions. Source data are available as a Source Data file

wildtype tau toxicity, referred to as Tau (high) transgenic *C. elegans*[20]. Compared to non-Tg animals, Tau (high) animals exhibit strong behavioral and neurodegenerative phenotypes driven by pathological tau protein[20], enabling us to readily detect potential improvement in tauopathy phenotypes. We crossed Tau (high) transgenic animals with transgenic animals expressing a pan-neuronal constitutively active *xbp-1s* gain of function

transgene (*xbp-1s* Tg), which promotes activation of the UPR^ER in the absence of increased ER protein folding load[12].

*C. elegans* expressing *xbp-1s* Tg have mild locomotion defects when compared to non-Tg animals (Supplementary Fig. 5), while Tau (high) *C. elegans* have severe locomotion dysfunction (Fig. 3a). However, Tau (high); *xbp-1s* Tg animals had robust rescue of the tau-mediated behavioral deficits compared to Tau (high)

animals alone (Fig. 3a). To test whether *xbp-1s* Tg promotes tau protein turnover, we examined tau levels by immunoblot. Tau (high); *xbp-1s* Tg animals accumulated less total tau protein when compared to Tau (high) animals (Fig. 3b, c). To address the possibility *xbp-1s* hyperactivity decreases tau protein levels by an effect on tau transgene expression rather than protein turnover, we assessed human tau mRNA levels by quantitative reverse-transcription PCR (qRT-PCR). We found no significant difference in tau-encoding mRNA (Supplementary Fig. 6), indicating reduced levels of tau protein are a result of increased tau protein clearance rather than decreased transgenic tau mRNA expression. In tauopathy disorders, tau becomes hyperphosphorylated, driving pathological protein aggregation and neurodegeneration[22]. We measured tau phosphorylation by immunoblot at four sites commonly associated with tauopathy (Threonine 181, Serine 202, Serine 396/404, and Serine 422) using phosphorylated tau-specific antibodies. Tau (high); *xbp-1s* Tg *C. elegans* had less phosphorylated tau at all four epitopes compared to Tau (high) animals (Fig. 3d, Supplementary Table 3). We then tested whether *xbp-1s* gain of function would protect against tau-driven neurodegeneration. At day 1 of adulthood, ~50% of Tau (high) *C. elegans* lost one GABAergic neuron (Fig. 3e, f). However, Tau (high); *xbp-1s* Tg worms were protected against this early tau-induced neuronal loss (Fig. 3e, f). By day 4 of adulthood, Tau (high) worms showed progressive neuronal loss with an average of almost three neurons lost per animal, while Tau (high); *xbp-1s* Tg worms that exhibited neurodegeneration lost approximately one neuron (Fig. 3g). Altogether, we have demonstrated *xbp-1s* gain of function significantly suppresses behavioral, biochemical, and neurodegenerative tauopathy phenotypes observed in the *C. elegans* Tau (high) background.

**ATF-6 mediates *xbp-1s* suppression of tauopathy.** The XBP-1s transcription factor mediates one of three branches of the UPR^ER (Fig. 1a). To understand whether the other two UPR^ER branches participate in *xbp-1s*-mediated tauopathy suppression in *C. elegans*, we again utilized *pek-1* and *atf-6* loss of function mutant strains. Tau (high); *xbp-1s* Tg animals were crossed with either *pek-1* or *atf-6* loss of function mutant strains.

We evaluated the consequences of deleting *pek-1* in Tau (high) animals alone and observed no change in tau-induced behavioral dysfunction (Fig. 4a), consistent with our data for Tau (low) animals (Fig. 1b). Next, we tested whether *pek-1* (−/−) could prevent *xbp-1s* suppression of tau toxicity. We found *pek-1* loss of function partially blocked the suppression of tau motility dysfunction in Tau (high); *xbp-1s* Tg; *pek-1* (−/−) animals (Fig. 4a). We then examined the effect of *atf-6* (−/−) on Tau (high) animals. In contrast to our previous finding in Tau (low) animals (Fig. 1c), we found Tau (high); *atf-6* (−/−) *C. elegans* had similar behavioral abnormalities when compared to Tau (high) animals (Fig. 4b). The severity of behavioral dysfunction of the Tau (high) animals likely explains our inability to discriminate enhancement of the tauopathy phenotype. Finally, we tested whether *atf-6* (−/−) prevents *xbp-1s*-mediated suppression of tau toxicity. Surprisingly, we found *atf-6* loss of function completely blocked *xbp-1s*-mediated tauopathy suppression in Tau (high); *xbp-1s* Tg; *atf-6* (−/−) animals (Fig. 4b). We confirmed these results with an independent *atf-6* loss of function allele (Supplementary Fig. 7a).

Because *xbp-1s*-mediated tauopathy behavioral suppression requires *atf-6*, we also measured *atf-6* loss of function effects on total tau protein by immunoblot in Tau (high) animals using two independent *atf-6* loss of function alleles. Loss of function of *atf-6* alone did not change tau protein levels in Tau (high) animals (Fig. 4c, d; Supplementary Fig. 7b). We have shown *xbp-1s* gain of

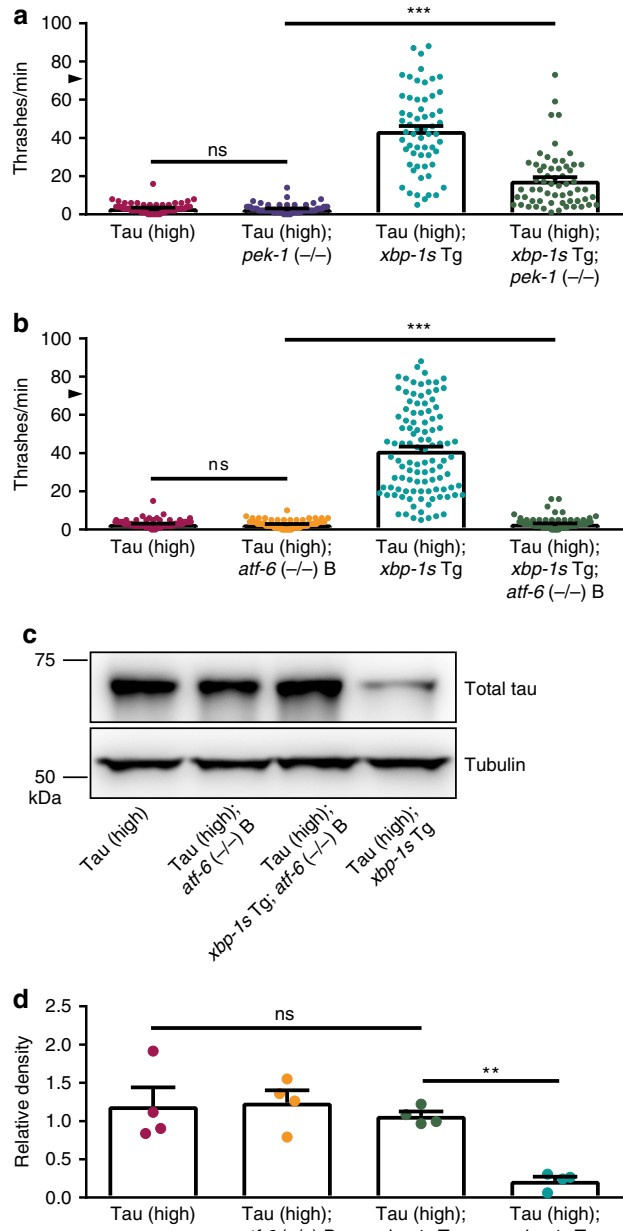

**Fig. 4** The UPR^ER branches are differentially involved in *xbp-1s*-mediated tauopathy suppression in transgenic *C. elegans*. **a** *pek-1* loss of function partially blocks the ability of neuronal overexpression of *xbp-1s* in Tau (high) animals to suppress severe behavioral defects observed in a liquid environment [$n = 60$ animals; $N = 4$ biologically independent experiments; statistical analysis is by one-way ANOVA, followed by Tukey's post-test (***$p < 0.0001$)]. **b** *atf-6* loss of function abolishes the ability of neuronal overexpression of *xbp-1s* in Tau (high) animals to suppress severe behavioral defects observed in a liquid environment [$n = 105$ animals; $N = 7$ biologically independent experiments; statistical analysis is by one-way ANOVA, followed by Tukey's post-test (***$p < 0.0001$)]. **c**, **d** *atf-6* loss of function abolishes the ability of neuronal overexpression of *xbp-1s* in Tau (high) animals to decrease soluble tau protein levels. Representative immunoblots for total tau and tubulin are shown, and densitometry analysis of chemiluminescence signals for tau normalized to tubulin are plotted [$N = 4$ biologically independent experiments; statistical analysis is by repeated measures ANOVA, followed by Tukey's post-test (**$p = 0.0019$)]. All bar graphs represent mean + SEM. Arrowhead on y-axis of all liquid thrashing bar graphs denotes non-Tg animals average ~70 thrashes/min under standard laboratory conditions. Source data are available as a Source Data file

function reduced tau accumulation in Tau (high) animals (Figs. 3b, c; 4c, d; Supplementary Fig. 7b). In contrast, Tau (high); xbp-1s Tg; atf-6 (−/−) animals accumulated total tau protein similar to the level seen in Tau (high) animals (Fig. 4c, d; Supplementary Fig. 7b), indicating functional ATF-6 is necessary for xbp-1s-mediated tauopathy detoxification. Taken together, these data demonstrate distinct, but overlapping roles for the three branches of the UPR^ER in regulating tau proteostasis in transgenic C. elegans.

**ERAD is required for *xbp-1s*-mediated tauopathy suppression.** Dysregulated proteostasis within the ER triggers the UPR^ER, resulting in activation of ER-associated protein degradation (ERAD). This provides a means for the direct removal of misfolded proteins inside the ER lumen and facilitates their degradation through either autophagy or the proteasome[23]. To determine whether tau protein resides within the ER, we crossed Tau (high) animals with transgenic animals expressing a fluorescently-tagged ER integral membrane protein, CP450::mCherry[24]. We co-immunostained transgenic animals with antibodies targeting tau and mCherry to visualize subcellular localization of these proteins (Fig. 5a–c). Using colocalization immunofluorescence analysis, we showed no significant overlap of tau protein and the ER protein CP450 (Fig. 5d). Thus, the mechanism of tau detoxification by UPR^ER activation in transgenic C. elegans does not rely upon removal of tau protein directly from the ER.

Previous work in cultured human cells demonstrated tau impairs ERAD function[25]. We hypothesized abnormal tau also inhibits ERAD in C. elegans, and neuronal tau toxicity in C. elegans may depend upon ERAD impairment. In support of this hypothesis, previous work has shown UPR^ER activation stimulates ERAD through XBP-1s target genes[26]. Therefore, XBP-1s suppression of tauopathy phenotypes in C. elegans may be through an ERAD-dependent mechanism. To test the involvement of ERAD in tau clearance, we examined whether ERAD loss of function could block XBP-1s suppression of tauopathy in C. elegans. SEL-11, the C. elegans homolog of the ERAD-associated E3 ubiquitin ligase HRD1, is required for ERAD[27]. We crossed sel-11 loss of function mutants [sel-11 (−/−)] with Tau (high); xbp-1s Tg animals and measured behavioral function. We found sel-11 (−/−) fully blocked XBP-1s-mediated suppression of tauopathy behavioral deficits (Fig. 5e). Taken together, these data suggest ERAD activation via XBP-1s promotes clearance of pathological tau in C. elegans.

## Discussion

Although aging both increases AD risk and decreases UPR^ER function, the specific contribution of UPR^ER activity to disease initiation and/or progression remains poorly understood. Likewise, whether UPR^ER induction causes or results from AD pathology remains unclear. Most studies to date have made correlative associations or relied upon immortalized cell culture experiments, which fail to model key aspects of tauopathy[28]. Here, we have taken a more direct approach by genetically dissecting the role of the UPR^ER in tauopathy in vivo using human wildtype tau transgenic C. elegans. We found xbp-1s expression to be a central control point in tauopathy, although this role requires involvement of all three branches of the UPR^ER.

To study the contributions of each UPR^ER branch, we systematically eliminated the function of the three UPR^ER branches in tau transgenic C. elegans. The PERK branch of the UPR^ER mainly functions to attenuate protein synthesis in the ER. Interestingly, we found C. elegans pek-1 does not regulate tau proteostasis on its own (Fig. 1b), indicating a non-essential role

for this branch in our tauopathy model. However, the PERK branch of the UPR^ER plays a modest role in modulating tauopathy risk, as human PERK variants increase risk for developing PSP by ~25%[29]. In addition, human post-mortem brain analyses found increased PERK activation in a subset of PSP[30] and AD disease-affected brain regions[30–32]. Our results indicate either the consequences of PEK-1 modulation are too modest to be detected in tau transgenic C. elegans, or there might be a critical species-specific difference between humans and C. elegans regarding PERK/PEK-1 function in the UPR^ER.

The XBP1 and ATF6 branches of the UPR^ER mediate transcriptional regulation of genes for restoring proteostasis in the ER. Our results show both atf-6 and xbp-1 loss of function enhance tauopathy-induced behavior defects in C. elegans (Figs. 1c, 2a; Supplementary Fig. 3), with xbp-1 loss of function additionally increasing tau protein levels and neurodegeneration (Fig. 2b–f). In humans, a polymorphism in the promoter of the XBP1 gene was found to be a risk factor for AD in a Chinese Han population[33]. Post-mortem tissue analyses have revealed a more complex scenario for XBP1 branch changes in AD, possibly related to disease stage, patient cohort characteristics, or experimental methodology. One group found the constitutively active form XBP1s is downregulated in AD brains[34]. Another group found increased XBP1s, but no increase in its key regulatory target BiP, indicating a disturbance in UPR^ER activation[35]. Work from our laboratory and others has demonstrated loss of UPR^ER function via XBP1 exacerbates pathological tau phenotypes in simple model organisms such as C. elegans ([19], this study) and Drosophila melanogaster (D. melanogaster)[36].

Several recent studies highlight the potential for the UPR^ER, specifically XBP1, as a therapeutic target for dementia. First, upregulation of XBP1s rescues Aβ-mediated neurotoxicity in D. melanogaster, mammalian cultured neurons, and C. elegans[37,38]. Second, Xbp1 gain of function in the hippocampus of mice can improve spatial memory[39]. To dissect the role of UPR^ER induction in protection from tauopathy, we expressed constitutively active xbp-1s in C. elegans neurons. We found XBP-1s protects neurons from pathological tau by promoting proteostasis and clearance of abnormal tau species, resulting in amelioration of both behavioral dysfunction and neurodegenerative changes in C. elegans (Fig. 3). Taken together, these data clearly support a critical role for UPR^ER transcriptional induction in mediating the clearance of abnormal protein species such as Aβ and tau, while shedding light on a novel active tau clearance mechanism in AD and other tauopathies.

Given the highly orchestrated nature of the UPR^ER (Fig. 1a), we next investigated whether the other two branches of the UPR^ER were genetically required for XBP-1s-mediated protection against pathological tau in C. elegans. Interestingly, this experiment identified interplay between the three branches of the UPR^ER. We observed C. elegans PEK-1 loss of function limits, but does not eliminate xbp-1s-mediated suppression of tauopathy-induced behavioral deficits (Fig. 4a). In C. elegans, the ire-1/xbp-1 and pek-1 branches play partially complimentary roles in eliminating ER stress, where ire-1/xbp-1 signals to activate UPR^ER transcription and pek-1 signals to attenuate protein synthesis[40]. In addition, pek-1 appears to be dispensable for the induction of genes directly involved in ER protein folding and ERAD[41]. The limited role of pek-1 in xbp-1s-mediated tauopathy suppression indicates a fully functioning UPR^ER is required to completely overcome cytoplasmic pathological tau protein accumulation in C. elegans. Nonetheless, transgenic overexpression of XBP-1s can partially suppress tauopathy even in the absence of pek-1 activity.

In contrast, we have shown atf-6 loss of function completely abolished xbp-1s-mediated tauopathy suppression in C. elegans (Fig. 4b–d; Supplementary Fig. 7). This demonstrates an absolute

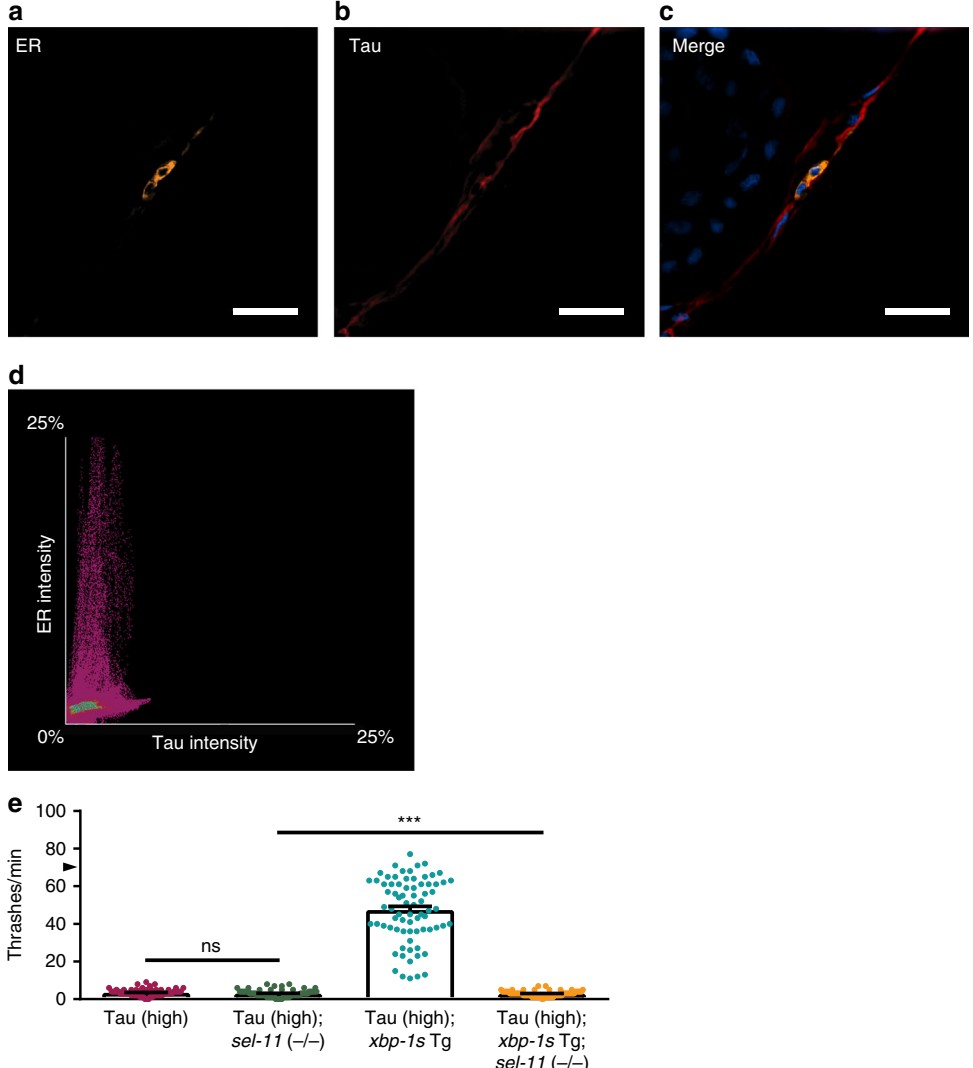

**Fig. 5** The ER does not contain tau protein, but ERAD is required for *xbp-1s*-mediated tauopathy suppression in transgenic *C. elegans*. **a–c** Tau protein does not colocalize with the ER in the Tau (high) *C. elegans* nervous system. Tau protein subcellular localization in relation to the ER integral membrane protein CP450 was measured by co-immunofluorescent staining and analyzed using structured illumination microscopy (SIM). **a** Representative SIM image of the ER protein CP450 fused to mCherry shows concentrated ER membrane localization primarily surrounding nuclei in the nerve cord [yellow = mCherry (ER); scale bar = 10 μm]. **b** Representative SIM image of tau protein shows diffuse tau localization throughout the nerve cord (red = tau; scale bar = 10 μm). **c** Representative merged SIM image of tau and ER membrane proteins, showing distinct subcellular localizations [yellow = mCherry (ER); red = tau; blue = DAPI (nuclei); scale bar = 10 μm]. **d** Colocalization analysis of ER reporter staining intensity (*y*-axis) with tau staining intensity (*x*-axis) shows no significant colocalization of the two proteins [statistical analysis is by Pearson correlation coefficient ($r = 0.23$)]. **e** *sel-11* loss of function abolishes the ability of neuronal overexpression of *xbp-1s* in Tau (high) animals to suppress severe behavioral defects observed in a liquid environment [$n = 75$ animals; $N = 5$ biologically independent experiments; bar graphs = mean + SEM; statistical analysis is by one-way ANOVA, followed by Tukey's post-test (***$p < 0.0001$)]. Arrowhead on *y*-axis denotes non-Tg animals average ~70 thrashes/min under standard laboratory conditions. Source data are available as a Source Data file

requirement for ATF-6 in mediating tauopathy suppression by transgenic constitutively active XBP-1s expression in *C. elegans* and a previously unappreciated critical crosstalk between ATF-6 and XBP-1 branches of the UPR[ER] in tau proteostasis. Given the known importance of the XBP-1 and ATF-6 branches in UPR[ER] transcriptional regulation, a plausible mechanism of tauopathy suppression involves regulation of specific target gene(s) coordinately upregulated by ATF-6n and XBP-1s transcription factors. In agreement with our results, previous studies suggest ATF-6n might have evolved as a backup mechanism to the XBP-1s transcriptional pathway in *C. elegans*, since many genes regulated by ATF-6n overlap with those regulated by XBP-1s[41]. In mammals, ATF6 activates the transcription of *XBP1* and can form heterodimers with XBP1s to control the induction of specific patterns of gene expression[42,43]. Additionally, mouse XBP1s/ATF6n heterodimers exhibit a higher binding affinity to target genes than XBP1s homodimers, indicating the strong influence of ATF-6n on XBP-1s transcriptional regulation[42]. Currently, it is unknown whether *C. elegans* XBP-1s and ATF-6n transcription factors can form heterodimers, but they do exhibit a high degree of sequence and functional conservation with mammalian XBP1 and ATF6 homologs. Taken together with the requirement for ATF-6 function in XBP-1s-mediated tauopathy suppression, we hypothesize these two transcription factors also form

heterodimers in *C. elegans* to promote tauopathy suppression. Alternatively, other scenarios involving an ordered pathway, where *atf-6* acts downstream of *xbp-1*, could explain the dependence of *xbp-1s* suppression on *atf-6*. Similarly, *atf-6* loss of function could cause synergistic deficits with pathological tau that *xbp-1s* overexpression cannot ameliorate. Regardless, the parallel findings for *atf-6* and *pek-1* loss of function suggests full UPR$^{ER}$ functionality promotes *xbp-1s*-mediated tauopathy suppression in *C. elegans*.

We found upregulating the UPR$^{ER}$ master transcription factor XBP-1s can increase tau protein turnover in the cytoplasm. Cytoplasmic tau can be degraded by at least two mechanisms: the UPS and autophagy. Soluble monomeric tau species are recognized by molecular chaperones and directed to the 20S proteasome without ubiquitination or 26S proteasome with ubiquitination for degradation, while aggregation-prone tau species, such as hyperphosphorylated tau, are degraded via autophagy[44,45]. Both of these protein degradation mechanisms are utilized by the UPR$^{ER}$ via ERAD, which shuttles proteins from the ER to the cytoplasm for degradation. Although tau does not reside in the ER, ERAD loss of function via *sel-11* (−/−) is essential for *xbp-1s*-mediated behavioral tauopathy suppression in *C. elegans* (Fig. 5), highlighting the importance of trans-compartmental trafficking mechanisms for global proteostasis.

We propose the following mechanism explaining the relationship between cytoplasmic tau and the UPR$^{ER}$ in neurons (Fig. 6). Although tau is not an ER resident protein, abnormal tau in the cytoplasm can nevertheless trigger the UPR$^{ER}$ by altering overall neuronal proteostasis through impaired ribosome function[46], ER/cytoplasmic protein translocation, and disruption of ERAD[25], leading to misfolded proteins in the ER. The main goal of the UPR$^{ER}$ is to restore proteostasis within the ER lumen, but to achieve this requires integrated stress relief within both cytoplasm and ER compartments. This is highlighted by a recent study demonstrating soluble tau impairs ERAD, which results in increased ER protein folding load and activation of the UPR$^{ER}$ [25] (Fig. 6a). Nascent proteins in the ER failing to properly fold are degraded through ERAD, a branch of the UPS in which proteins are ubiquitinated at the ER membrane for subsequent degradation. Although this is an ER-associated degradation pathway, the ERAD machinery can degrade abnormal proteins located outside the ER, such as in the cytoplasm or nucleus[47]. XBP-1s UPR$^{ER}$ branch loss of function inhibits ERAD, leading to pathological accumulation of abnormal tau as it is converted from soluble normal tau in the cytoplasm (Fig. 6b). In the presence of pathological cytoplasmic tau, XBP-1s UPR$^{ER}$ branch gain of function stimulates high levels of ERAD, leading to turnover of pathological tau species, such as hyperphosphorylated tau (Fig. 6b). Taken together with our findings, these studies indicate the importance of understanding the role of abnormal tau protein and the UPR$^{ER}$ in age-associated neurodegenerative disease.

Future translational studies will need to address whether UPR$^{ER}$ activation can protect against pathological tau in the mammalian brain. More broadly, exploring how different cellular compartments communicate to achieve neuronal proteostasis remains an important mechanistic question with therapeutic implications. Specifically, the mechanism of how cytoplasmic pathological tau feeds back on ER folding load needs to be identified. Given the importance of the two transcriptional branches of the UPR$^{ER}$ in tauopathy, our next step will be to identify *xbp-1s* target genes driving tau proteostasis in *C. elegans*, which could help inform more specific neuroprotective strategies in mammals. Likewise, involvement of the UPR$^{ER}$ in tauopathy needs to be explored in the mammalian brain to clarify whether targeting the UPR$^{ER}$ in tauopathy could have neuroprotective benefit.

## Methods

**C. elegans strains and transgenics.** *C. elegans* strains used are listed in Supplementary Table 1. All strains were maintained at 20 °C on standard nematode growth media plates containing OP50 *Escherichia coli* (*E. coli*)[48]. *C. elegans* were grown on nematode growth media plates containing five times more peptone (5X PEP) prior to collection for protein and RNA extraction. The *C. elegans* husbandry and experimentation was conducted in accordance with all relevant ethical and safety regulations for animal testing and research.

**C. elegans behavioral analysis.** *C. elegans* swimming behavior was performed as described previously[18]. *C. elegans* were developmentally synchronized using a timed egg-lay and grown until all reached day one of adulthood at 20 °C. Individual developmentally synchronized animals were placed in a 10 μL M9 buffer droplet on a 10-well Teflon-printed glass slide and allowed to acclimate to a liquid environment for 10 s. One thrash was defined as a bend across the midline or two consecutive bends from the midline toward the same side. The number of thrashes were counted during a 1-min period and averaged for each strain. At least three biological replicate assay sessions with at least 10 animals per assay session were analyzed for statistical significance. Behavioral analysis was conducted by S.M.W., who was blinded to genotype to avoid biases. However, when the strains to be assayed were visibly distinguishable, blinding was not feasible. For comparison of two groups, an unpaired *t*-test, two-tailed, was used. For comparisons of three or more groups, a one-way analysis of variance (ANOVA), followed by Tukey's multiple comparison post-test was used.

**C. elegans neurodegeneration analysis.** Analysis of *C. elegans* neurodegeneration was performed using a transgenic reporter strain with fluorescently marked VD and DD type inhibitory motor neurons[18]. *C. elegans* were crossed with the EG1285 strain (*unc-47p::gfp*)[21] to visualize the ventral nerve cord green fluorescent protein (GFP) marked gamma-aminobutyric acid (GABA)ergic neurons via fluorescence microscopy. *C. elegans* were developmentally synchronized using a timed egg-lay and grown until all reached day one or four of adulthood. *C. elegans* were mounted on glass slides with a 2% agarose gel pad and 10 μL 0.1% sodium azide. The number of intact ventral nerve cord GABAergic D-type motor neurons were counted under fluorescence microscopy on a DeltaVision Elite (GE, Issaquah, WA, USA) imaging system using an Olympus ×60 oil objective. Representative images of neuronal loss were also taken using an Olympus ×60 oil objective. The three most rostral GABAergic motor neurons were excluded from counts due to occlusion from pharyngeal GFP co-injection markers. Therefore, the number of visibly accessible ventral nerve cord D-type motor neurons were counted out of a total of 16. Neurodegeneration was plotted as the average number of neurons lost from the 16 possible neurons analyzed. Two biological replicate assay sessions with at least 10 animals per assay session were analyzed for statistical significance. For comparison of two groups, an unpaired *t*-test, two-tailed, was used. Representative anatomical images were created with ImageJ Java[49] to overlay images from the POL and GFP channels, with the GFP channel image re-colored green and the POL channel image re-colored grey for emphasis. Images from the GFP channel were also shown below merged images for visual clarity of neuronal loss.

**C. elegans protein extraction.** Protein was extracted from *C. elegans* using a rigorous lysis approach[18]. *C. elegans* were synchronized using hypochlorite treatment and grown until adulthood at 20 °C on 5X PEP plates. Synchronized adult *C. elegans* were washed off 5X PEP plates in M9 buffer, washed an additional four times in M9 buffer to remove excess OP50 *E. coli*, pelleted by a 1-min centrifugation at 800 × *g*, snap frozen in liquid nitrogen, and stored at −70 °C until protein extraction. A total of 2 μL of high-salt reassembly (RAB) buffer [0.1 M 2-(*N*-morpholino)ethanesulfonic acid, 1 mM ethylene glycol bis-2-aminoethyl ether-N,N',N'',n'-tetraacetic acid, 0.5 mM MgSO$_4$, 0.75 M NaCl, 0.02 M NaF, pH 7.0] containing phenylmethylsufonyl fluoride and protease inhibitors was added per milligram of packed worm pellet wet weight and homogenized via sonication four times at 70% power for 8 s. This total soluble worm lysate was reserved for subsequent immunoblotting.

**Protein immunoblotting.** *C. elegans* protein immunoblotting was performed using Criterion apparatus (Bio-Rad) as recommended by the manufacturer (Bio-Rad Laboratories, Hercules, CA, USA)[18]. *C. elegans* protein preparations were diluted 5:1 with sample buffer (0.046 M Tris, 0.005 M ethylenediaine tetraacetate, 0.2 M dithiothreitol, 50% sucrose, 5% sodium dodecyl sulfate, 0.05% bromophenol blue), boiled for 10 min, and centrifuged at 16,100 × *g* for 2 min. A total of 10 μL of diluted protein preparations were loaded and resolved on precast 4–15% gradient sodium dodecyl sulfate polyacrylamide (SDS-PAGE) gel and transferred to polyvinylidene difluoride (PVDF) membrane as recommended by the manufacturer (Bio-Rad Laboratories, Hercules, CA, USA). PVDF membranes were blocked in 5% milk in phosphate-buffered saline (PBS) for 1 h before overnight incubation with primary antibody at 4 °C. The next day, PVDF membranes were washed in PBS with 0.05% Tween, incubated at room temperature with horseradish peroxidase (HRP)-coupled secondary antibody for 2 h, and washed in PBS with 0.05% Tween before visualization. The dilutions and pertinent details for all primary and secondary antibodies used are listed in Supplementary Table 2. Enhanced

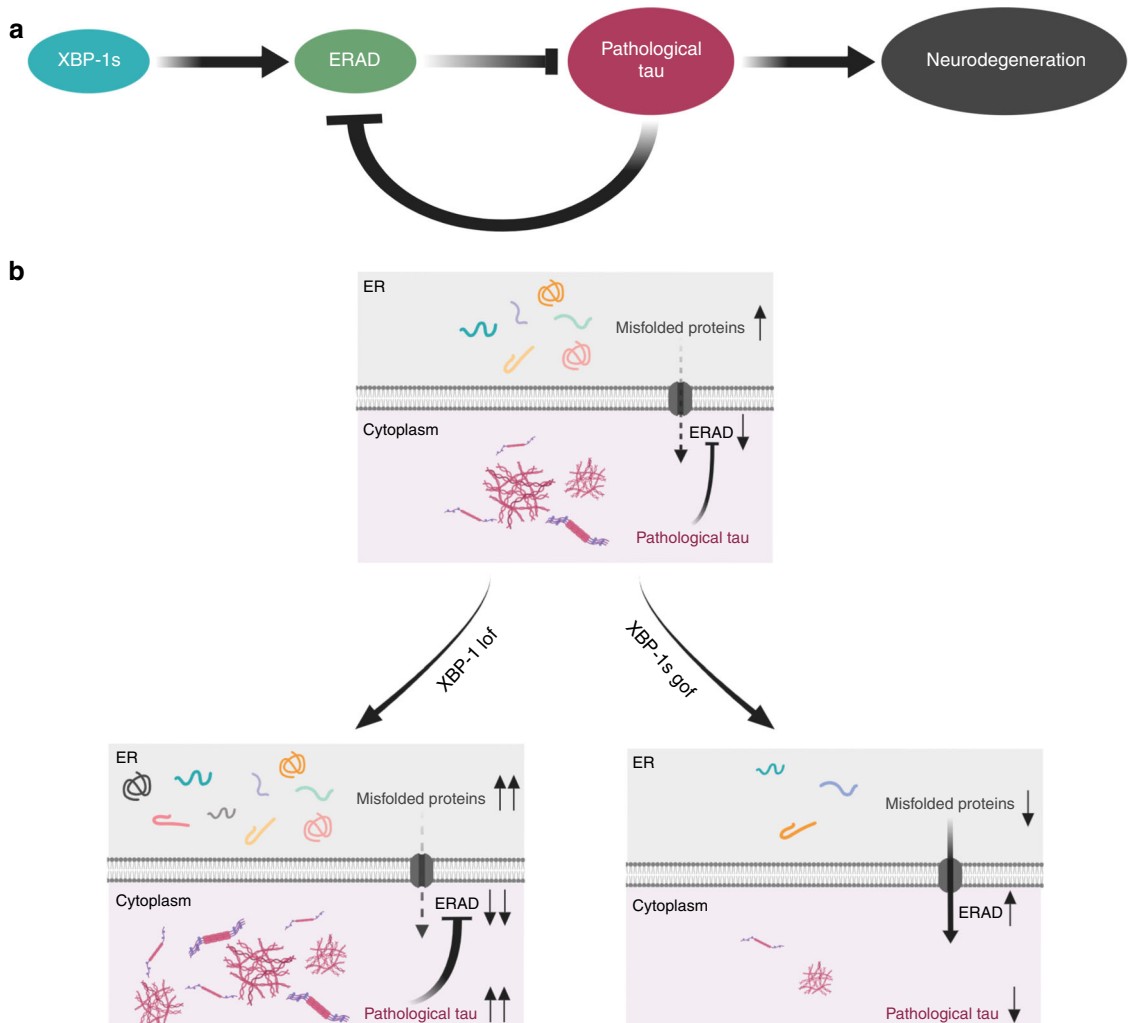

**Fig. 6** Activation of the UPR^ER XBP-1 transcriptional branch mediates detoxification of pathological tau via ERAD. **a** The UPR^ER master transcription factor XBP-1s controls transcription of ERAD genes. ERAD degrades proteins via the proteasome and autophagy, and ERAD loss of function can inhibit degradation of pathological tau. The presence of pathological tau in the cytoplasm is a stressor that impairs ERAD, leading to the accumulation of ER resident misfolded proteins and induction of the UPR^ER25. **b** Loss of the UPR^ER XBP-1 branch promotes conversion of non-pathological tau to pathological tau, potentially by inhibition of ERAD, resulting in toxicity and subsequent neuronal cell death in *C. elegans*. Constitutive activation of the UPR^ER master transcription factor XBP-1s stimulates ERAD and promotes clearance of pathological tau, leading to reduction of toxicity and subsequent neuronal cell survival in *C. elegans*

chemiluminescence substrate (Bio-Rad Laboratories, Hercules, CA, USA) was added to the PVDF membrane, and chemiluminescence signals were detected with ChemiDoc-It[®2 510 Imager (UVP LLC, Upland, CA, USA). Relative intensity of chemiluminescence signals was measured with ImageJ Java[49]. At least three biological assay replicates were analyzed for statistical significance. For comparison of two groups with biological replicates run on independent or single immunoblots, a paired *t*-test, two-tailed, was used. For comparisons of three or more groups run on independent or single immunoblots, a repeated measures analysis of variance (ANOVA), followed by Tukey's multiple comparison post-test was used. Full-length immunoblot images are available for all experiments as a Source Data file called: Immunblots.

**C. elegans freeze cracking for immunofluorescent staining.** *C. elegans* immunofluorescent staining was performed according to standard protocols as described by Janet Duerr[18,50]. Tau (high) animals were crossed with the CX10344 kyEx2454 strain (*unc-25::calf-1::gfp, unc-25::CP450::mCherry, odr-1::dsRED*)[24] to visualize the subcellular localization of tau protein relative to the endoplasmic reticulum via fluorescence microscopy. *C. elegans* were developmentally synchronized using a timed egg-lay and grown until all reached day one of adulthood at 20 °C. Individual developmentally synchronized animals were placed in a 20 μL M9 buffer droplet on a 10-well Teflon-printed glass slide and washed to remove excess OP50 *E. coli* by moving to successive 20 μL M9 buffer droplets three times. Washed animals were transferred to a 10 μL 1% paraformaldehyde solution droplet on a poly-L-lysine coated glass slide and freeze cracked[50]. Adherent animals were

exposed to a light fixation using methanol-acetone for 5 min each at −20 °C. Freeze cracked, fixed animals were stained and washed as described. Animals were counterstained with 300 nM DAPI and mounted with ProLong™ Gold antifade (Invitrogen, Carlsbad, CA, USA) before super-resolution microscopy analysis. The dilutions and pertinent details for all primary and secondary antibodies used are listed in Supplementary Table 2. Three-dimensional structured illumination microscopy (3D-SIM) images of animals were acquired on a Nikon N-SIM-E Ti2 microscope (Tokyo, Japan) using a CFI SR HP Apochromat TIRF 100XC Oil (NA 1.49) objective, 561/647 nm diode lasers, and an ORCA-Flash 4.0 sCMOS camera (Hamamatsu Photonics K.K., Naka-ku, Hamamatsu City, Shizuoka, Japan). Image reconstruction and colocalization analysis was conducted using NIS-Elements (Nikon, Minato, Tokyo, Japan).

**C. elegans RNA extraction.** RNA was purified from snap-frozen packed *C. elegans* pellets using TRIzol Reagent as directed by the manufacturer's instructions[51]. RNA was resuspended in 50 uL sterile water. RNA concentration and purity were assessed using a NanoPhotometer® NP80 spectrophotometer (Implen GmbH, Munich, Germany). RNA integrity was assessed by 1% Tris/Borate/EDTA (TBE) agarose gel electrophoresis.

**Quantitative RT-PCR.** cDNA was prepared using iScript™ Reverse Transcription Supermix for RT-qPCR (Bio-Rad Laboratories, Inc., Hercules, CA, USA) as recommended by the manufacturer[51]. The primer set used to measure human

*MAPT* gene was (forward primer: 5′-GTGTGGCTCATTAGGCAACATCC-3′, reverse primer: 5′-CGTTCTCGCGGAAGGTCAG-3′). The primer set used for normalization was designed to detect *C. elegans rpl-32* gene (forward primer: 5′-GGTCGTCAAGAAGAAGCTCACCAA-3′, reverse primer: 5′-TCTGCGGA-CACGGTTATCAATTCC-3′). Each genotype was tested with three biological replicates (including three technical replicates within each experiment) for each primer set. qPCR was performed using the iTaq™ Universal SYBR® Green Supermix kit (Bio-Rad Laboratories, Inc., Hercules, CA, USA) in a 384-well plate on a 7900HT Fast Real-Time PCR System (Applied Biosystems, Foster City, CA, USA). Data were normalized within samples using an internal reference control gene (*rpl-32*).

**Statistical analysis**. Statistical significance for all assays was determined using GraphPad Prism statistical software (GraphPad Software, Inc., La Jolla, CA, USA). Statistical significance is demarcated in figures as $*p < 0.05$, $**p < 0.005$, and $***p < 0.0001$.

**Schematic illustrations**. Original schematic illustrations were created using an academic license with the online application BioRender (Toronto, ON, CA).

**Reporting summary**. Further information on research design is available in the Nature Research Reporting Summary linked to this article.

## Data availability
The data analyzed for this study are published in this manuscript and associated supplementary information. The source data underlying Figs. 1, 2, 3, 4, and 5 and Supplementary Figs. 1, 2, 3, 4, 5, 6, and 7 are available as Source Data files. The source data underlying Figs. 1d, 2b, 3b and d, 4c; and Supplementary Fig. 7b are available as a Source Data file called: Immunoblots.

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

## Acknowledgements

We thank Aleen Saxton, Kaili Chickering, Jennifer Geist, and Ashley Yeung for essential technical assistance. We thank Aleen Saxton, Rebecca Kow, Elaine Loomis, Jeanna Wheeler, and Sarah Benbow for sharing their expertise. We thank Rebecca Taylor and Andrew Dillin for providing *xbp-1s* transgenic *C. elegans* strains and advice about UPR activation in *C. elegans*. We also thank Rebecca Taylor for constructive feedback on the manuscript. We thank Cornelia Bargmann for providing the ER reporter transgenic *C. elegans* strain. We thank the National BioResource Project (Japan) and *C. elegans* Genetics Center (CGC) for providing strains. We thank Mike Crowder (University of Washington) for providing the tm1153 strain. We thank the Developmental Studies Hybridoma Bank (NICHD) for providing the β-tubulin primary antibody E7. We thank Peter Davies (Feinstein Institute of Medical Research) for providing the CP13 and PHF-1 tau phosphorylation epitope primary antibodies. We thank Rebecca Hull and Daryl Hackney at the Cellular and Molecular Imaging Core of the University of Washington Diabetes Research Center (National Institutes of Health P30-DK017047 grant) for their assistance with N-SIM imaging. We thank the National Institutes of Health (R21-NS0903771 grant to B.C.K.), the National Institute on Aging (T32-AG052354 grant to S.M.W.), and the Department of Veterans Affairs (Merit Review Grant #I01BX002619 to B.C.K. and Merit Review Grant #I01BX004044 to N.F.L.) for funding and training support.

## Author contributions

S.M.W. performed experiments, analyzed data, and wrote the manuscript. T.J.S performed experiments and analyzed data. T.A.V. performed experiments. N.F.L. analyzed data and wrote the manuscript. B.C.K. conceived and oversaw the study, analyzed data, and wrote the manuscript. All authors contributed to the final version of the manuscript.

## Additional information

**Competing interests:** The authors declare no competing interests.

