## [Peer Review File · Nature Communications]

Reviewers' comments:

Reviewer #1 (Remarks to the Author):

In their manuscript, Waldherr et al. use a genetic dissection of a disease model system to present a compelling case that the unfolded protein response of the endoplasmic reticulum can modulate the aggregation and toxicity phenotypes of human tau expression in the neurons of *C. elegans*. They show that a mutant for the UPRER effector *xbp-1s* exhibits exacerbation in tau protein accumulation, neuronal degeneration, and the associated motor phenotype of thrashing. Furthermore, the authors demonstrate that overexpression of *xbp-1s* in neurons is able to rescue tau accumulation, degeneration, and the thrashing phenotype. Additionally, the authors explore dependence of this rescue on additional UPRER branches, finding that mutants for two parallel effectors, *pek-1* (aka PERK) and *atf-6*, cause a partial and full suppression of the rescue, respectively.

This manuscript exhibits a thoughtful genetic dissection of an otherwise opaque disease system, and the authors carefully and clearly develop a novel relationship between *xbp-1* and tau phenotypes. Only minor experimental updates are needed to support the conclusions the authors make in the body of the paper, and the findings are generally strong. Additional mechanistic details could always strengthen the work but seem to fall outside the scope of this story. However, a less speculative and more data-driven discussion that focuses on the novelty of the current study would strengthen the paper by extrapolating more on the present data than assertions from the literature.

General experimental comments are as follows:

- 1) It would be helpful to include control/non-Tg/N2 worms in the data visualizations for thrashing assays and for quantification of neurons lost with age, including next to the tau strains in the main figures. The comparison has value, especially when looking at rescue/suppression of rescue in the relevant manipulations.
- 2) The manuscript would benefit from analysis of some of the tau-associated phenotypes in the *xbp-1s* mutant by itself. Do these worms have a thrashing phenotype? Do they exhibit a difference in age-associated degradation of *unc-47+* neurons? These data would also be nice to see for the *xbp-1s* overexpresser, *atf-6*, and *pek-1*, if possible. It is possible that mutants for the more "terminal" branches of the UPRER could be protected against process degradation etc., or that the overexpressing worms are protected by themselves.

Model/Discussion-based comments:

- 1) Figure 5 centers almost entirely on ERAD, but no experiments specifically interrogating ERAD as a mechanism have been done in the paper. The authors ought to either include an ERAD measure or focus less on this mechanism in a major figure. Furthermore, by focusing on ERAD exclusively in the discussion, the authors either omit or only briefly mention other interesting possibilities. Discussion of ERAD of a potential mechanism is still relevant but needs to be dialed back somewhat.
- 2) Similarly, the analysis of the *atf-6* and *xbp-1* interaction data in the discussion purports the likelihood of a physical interaction between *xbp-1* and *atf-6*, although none of the data imply that this is more likely than any other mechanism. For example, *atf-6* knockdown could cause a compensatory increase in *xbp-1* and *pek-1* mediated processing, such that additional *xbp-1* is no longer able to rescue beyond the level needed to keep *atf-6* mutants alive. Activation of *atf-6* could also occur downstream of *xbp-1s* but might be required for expression of tau for some reason. Additionally, this neuronal *xbp-1s* overexpressing strain is supposed to invoke UPRER in multiple cell types- could these other cells have a protective function feeding back on neurons that is somehow dependent on *atf-6*? None of these questions seem to be within the scope of this study, but they rely on the present data to speculate a number of possible mechanisms.

Overall, this manuscript is a broadly interesting and highly novel genetic dissection of interesting disease biology. It capitalizes on the strengths of multiple fields and a highly genetically tractable

system to pull apart interactions between tauopathies and UPRER that are almost entirely unstudied, and rarely in such a compelling manner. We recommend this manuscript's publication with the above minor revisions.

Reviewer #2 (Remarks to the Author):

This is an interesting and relevant study demonstrating a clear role for XBP-1 in the turnover of tau in a *C. elegans* tauopathy model. The ER UPR has been implicated in multiple neurodegenerative conditions, and this study convincingly shows that it modulates tau levels in this *C. elegans* model. These results are novel, and the methods and analysis are presented in sufficient detail to replicate the findings. I have one major and a few minor points with respect to this manuscript:

Major point

Significant text is employed in the Discussion to explain how the presumably cytoplasmic tau protein is degraded in an ER-UPR-dependent manor. This raises two issues: 1) is human tau really cytoplasmic in this model in the first place, and if so 2) what protein degradation mechanism is employed in XBP-1-dependent tau turnover? While it seems unlikely tau ends up in the ER in this model, I do not think any of the previous studies by this group has determined the subcellular localization of tau in the transgenic models. While a high-resolution immunofluorescence image of tau staining along with an ER marker would be nice, at least showing that tau expression did not up-regulate hsp-4 or other ER stress markers would help settle this issue. Assuming tau is cytoplasmic, presumably its XBP-1 degradation depends on ERAD/proteasomal function and/or autophagy. Both these processes can be readily inhibited using appropriate mutations or RNAi knockdowns, and this study would be significantly stronger if these approaches were employed to determine the mechanism of XBP-1-dependent tau degradation.

Minor points

- 1) The manuscript should clarify up front that the tau models employed use non-mutant 4R tau.
- 2) Although the brightfield/epifluorescence overlay images in Figures 2D and 3E nicely show the anatomical localization of the neurons being scored, it does make the determining whether there are really discontinuities in the axons difficult. I would supplement the overlay images with the straight epifluorescence images to make this point more apparent.
- 3) All the tau blots show only the range of the gel that covers full-length tau. Although this is reasonable for presentation purposes, it does obscure any additional information that could be inferred from the blot. Specifically, I could imaging a caspase cleavage of tau leading to the dramatic reduction shown for full-length tau in the XBP-1s, which would be apparent in lower MW bands on the gel. Representative full-length immunoblots should be included in the supplement.
- 4) The second paragraph of the discussion states: "... UPRER mainly functions to attenuate protein synthesis in the ER". If protein synthesis happens in the ER, I have been misleading my students for many years!
- 5) There is an obvious control not presented that hopefully has already been done: a demonstration that XBP-1s does not alter transcription of the tau transgene. It is not far-fetched that the *aex-3* promoter could be influenced by the XBP-1 transcription factor. While I am listing this as a minor point, this experiment needs to be presented.
- 6) It would be helpful if the resubmission included page numbers.

Christopher D. Link

Reviewer #3 (Remarks to the Author):

While it has been convincingly demonstrated that ER stress and the unfolded protein response are activated in Alzheimer's disease and other tauopathies, the details of this relationship have remained mostly unknown. A similar relationship has also been described for other neurodegenerative diseases. Thus, this study by Waldherr et al. that seeks to evaluate if one or more of the branches of the UPR within the ER can offer protection against tau-mediated toxicity and associated phenotypes is highly relevant and the findings are potentially broadly applicable across many diseases hallmarked by protein aggregates. The authors used a genetic approach in *C. elegans* to systematically delete the regulators of this feedback system independently or in combination to better understand their connection to tau pathogenesis. They evaluated the outcomes by measuring motor behavior, neuronal counts, and Western blots. Overall, this work has revealed a very interesting and important finding that XBP1 significantly contributes to tau pathogenesis, in that in the absence of this gene tau toxicity and related phenotypes are worsened and in the presence of a constitutively active splice variant of XBP1 there is protection. The data also clearly show a role for the ATF6 and PERK branches in regulating tau pathogenesis through XBP1s. However, when silenced alone, PERK (PEK) deletion had no effects on tau-LOW phenotypes, while ablation of ATF6 worsened tau-associated phenotypes without affecting total tau levels. Overall, this is a fairly thorough study that supports the authors' conclusions. I only have a few comments that should be addressed.

1. Quantification should be shown for all Western blots, including those in the supplement.
 - a. Figure 3D would be more informative with a pTau/total tau ratio shown to determine any specific pTau changes
2. The authors suggest that the effects are through altered ERAD. A measurement of ERAD function should be provided to support these claims.
3. Further discussion about the discrepancy between the XBP1s behavioral defect and the XBP1s/high tau behavioral rescue should be included.
4. Even though it is outside the scope of the current study to investigate, the inclusion of a discussion of what genes are activated by XBP1s constitutive activation that could regulate tau turnover would be beneficial.

Reviewer #1 (Reviewer's remarks are in italics; Response by the Authors are in regular type):

In their manuscript, Waldherr et al. use a genetic dissection of a disease model system to present a compelling case that the unfolded protein response of the endoplasmic reticulum can modulate the aggregation and toxicity phenotypes of human tau expression in the neurons of C. elegans. They show that a mutant for the UPRER effector xbp-1s exhibits exacerbation in tau protein accumulation, neuronal degeneration, and the associated motor phenotype of thrashing. Furthermore, the authors demonstrate that overexpression of xbp-1s in neurons is able to rescue tau accumulation, degeneration, and the thrashing phenotype. Additionally, the authors explore dependence of this rescue on additional UPRER branches, finding that mutants for two parallel effectors, pek-1 (aka PERK) and atf-6, cause a partial and full suppression of the rescue, respectively.

This manuscript exhibits a thoughtful genetic dissection of an otherwise opaque disease system, and the authors carefully and clearly develop a novel relationship between xbp-1 and tau phenotypes. Only minor experimental updates are needed to support the conclusions the authors make in the body of the paper, and the findings are generally strong. Additional mechanistic details could always strengthen the work but seem to fall outside the scope of this story. However, a less speculative and more data-driven discussion that focuses on the novelty of the current study would strengthen the paper by extrapolating more on the present data than assertions from the literature.

General experimental comments are as follows:

1) It would be helpful to include control/non-Tg/N2 worms in the data visualizations for thrashing assays and for quantification of neurons lost with age, including next to the tau strains in the main figures. The comparison has value, especially when looking at rescue/suppression of rescue in the relevant manipulations.

We appreciate the Reviewer's point. For behavioral data, we now provide this information in the main figures for non-Tg animals by indicating wildtype motility levels (~70 thrashes/minute) with an arrowhead on the y-axis of all behavioral data graphs (see Fig. 1b, 1c, 2a, 3a, 4a, 4b, 5e; Supplementary Fig. 3, 7a). We also include thrashing data comparing non-Tg to *pek-1* (-/-), *atf-6* (-/-), *xbp-1* (-/-), and *xbp-1s* Tg (Supplementary Fig. 1a, 1b, 2, 5, respectively). For neurodegeneration data, we now provide this information in the main figures for non-Tg animals with an arrowhead on the y-axis indicating non-Tg animals do not exhibit neuronal loss (Fig. 2e, 3f). We also include data demonstrating no significant neuronal loss in non-Tg, *xbp-1* (-/-), and *xbp-1s* Tg animals (Supplementary Fig. 4). Note that we now make all behavioral and neurodegeneration data graphs internally consistent with identical scaling.

2) The manuscript would benefit from analysis of some of the tau-associated phenotypes in the xbp-1s mutant by itself. Do these worms have a thrashing phenotype? Do they exhibit a difference in age-associated degradation of unc-47+ neurons? These data would also be nice to see for the xbp-1s overexpresser, atf-6, and pek-1, if possible. It is possible that mutants for the more "terminal" branches of the UPRER could be protected against process degradation etc., or that the overexpressing worms are protected by themselves.

We now include data showing an absence of neurodegeneration in non-Tg control animals, *xbp-1* (-/-), and *xbp-1s* Tg (Supplementary Fig. 4). We see no obvious changes in neuron morphology or degeneration in the absence of tau. Due to resubmission time constraints, we were unable to address whether *atf-6* (-/-) and *pek-1* (-/-) affected neurodegeneration with age in the absence of tau. However, we have included complete behavioral datasets for these control strains demonstrating no significant differences in locomotion compared to non-Tg animals (Supplementary Fig. 1). Because neither *atf-6* (-/-) nor *pek-1* (-/-) significantly modified tau-dependent locomotion phenotypes (Fig. 1b, 1c), it is likely there are no significant effects on neurodegeneration as behavioral impairment typically accompanies neuronal loss in tau transgenic strains.

Model/Discussion-based comments:

1) Figure 5 centers almost entirely on ERAD, but no experiments specifically interrogating ERAD as a mechanism have been done in the paper. The authors ought to either include an ERAD measure or focus less on

this mechanism in a major figure. Furthermore, by focusing on ERAD exclusively in the discussion, the authors either omit or only briefly mention other interesting possibilities. Discussion of ERAD of a potential mechanism is still relevant but needs to be dialed back somewhat.

We agree and now provide additional experimentation in support of the model. Using genetic loss of ERAD function mediated by *sel-11* (-/-), we demonstrate XBP-1's suppression of tauopathy requires a functional ERAD pathway (see Fig. 5e).

*2) Similarly, the analysis of the *atf-6* and *xbp-1* interaction data in the discussion purports the likelihood of a physical interaction between *xbp-1* and *atf-6*, although none of the data imply that this is more likely than any other mechanism. For example, *atf-6* knockdown could cause a compensatory increase in *xbp-1* and *pek-1* mediated processing, such that additional *xbp-1* is no longer able to rescue beyond the level needed to keep *atf-6* mutants alive. Activation of *atf-6* could also occur downstream of *xbp-1*s but might be required for expression of tau for some reason. Additionally, this neuronal *xbp-1*s overexpressing strain is supposed to invoke UPRER in multiple cell types- could these other cells have a protective function feeding back on neurons that is somehow dependent on *atf-6*? None of these questions seem to be within the scope of this study, but they rely on the present data to speculate a number of possible mechanisms.*

The reviewers point is well taken. We appreciate that our data does not directly speak to this question and acknowledge it is beyond the scope of the present study. To incorporate a broader perspective on this topic, we have modified our discussion to include some of these possibilities and eliminated the reliance on the existence of an *xbp-1*s and *atf-6* heterodimer as a regulatory mechanism in our data summary figure (Fig. 6).

Overall, this manuscript is a broadly interesting and highly novel genetic dissection of interesting disease biology. It capitalizes on the strengths of multiple fields and a highly genetically tractable system to pull apart interactions between tauopathies and UPRER that are almost entirely unstudied, and rarely in such a compelling manner. We recommend this manuscript's publication with the above minor revisions.

We appreciate Reviewer 1's enthusiasm for the work and hope our revisions satisfy their previous concerns.

Reviewer #2 (Reviewer's remarks are in italics; Response by the Authors are in regular type):

This is an interesting and relevant study demonstrating a clear role for XBP-1 in the turnover of tau in a C. elegans tauopathy model. The ER UPR has been implicated in multiple neurodegenerative conditions, and this study convincingly shows that it modulates tau levels in this C. elegans model. These results are novel, and the methods and analysis are presented in sufficient detail to replicate the findings. I have one major and a few minor points with respect to this manuscript:

Major point

*Significant text is employed in the Discussion to explain how the presumably cytoplasmic tau protein is degraded in an ER-UPR-dependent manner. This raises two issues: 1) is human tau really cytoplasmic in this model in the first place, and if so 2) what protein degradation mechanism is employed in XBP-1-dependent tau turnover? While it seems unlikely tau ends up in the ER in this model, I do not think any of the previous studies by this group has determined the subcellular localization of tau in the transgenic models. While a high-resolution immunofluorescence image of tau staining along with an ER marker would be nice, at least showing that tau expression did not up-regulate *hsp-4* or other ER stress markers would help settle this issue. Assuming tau is cytoplasmic, presumably its XBP-1 degradation depends on ERAD/proteasomal function and/or autophagy. Both these processes can be readily inhibited using appropriate mutations or RNAi knockdowns, and this study would be significantly stronger if these approaches were employed to determine the mechanism of XBP-1-dependent tau degradation.*

We appreciate the Reviewer's concerns regarding whether tau is mislocalized to the ER, and to address the Reviewer's point "1)" above, we conducted the requested co-immunofluorescent analysis to

determine the relationship between tau and ER colocalization. We see essentially no overlap between tau and ER (see Fig. 5a-d) by co-immunofluorescent Structured Illumination Microscopy (SIM). We share a common interest with the Reviewer regarding the mechanisms of tau turnover in this model system. Our specific efforts to address point "2)" have been ongoing for some time, but we have previously not been able to demonstrate a specific requirement of proteasomal function in tau Tg animals [see Kraemer, et al 2003, Kraemer et al 2006, Guthrie et al, 2009].. However, we have added new data showing ERAD is required for XBP-1s suppression of tau-related behavioral phenotypes (see Fig. 5e, and our response to Reviewer 1 regarding the model/discussion-based comments, point "1)", above addressing a similar issue).

Minor points

1) The manuscript should clarify up front that the tau models employed use non-mutant 4R tau.

We now explicitly describe this in the Introduction, Methods, Results, and Discussion (see page 5, line 102 of Introduction, page 6, line 116 of Methods, page 10, line 250 of Results).

2) Although the brightfield/epifluorescence overlay images in Figures 2D and 3E nicely show the anatomical localization of the neurons being scored, it does make the determining whether there are really discontinuities in the axons difficult. I would supplement the overlay images with the straight epifluorescence images to make this point more apparent.

We now provide epifluorescence images below the overlay panels in the main figures (Fig. 2d, 3e).

3) All the tau blots show only the range of the gel that covers full-length tau. Although this is reasonable for presentation purposes, it does obscure any additional information that could be inferred from the blot. Specifically, I could imagine a caspase cleavage of tau leading to the dramatic reduction shown for full-length tau in the XBP-1s, which would be apparent in lower MW bands on the gel. Representative full-length immunoblots should be included in the supplement.

We now include the full-length Western blots for all immunoblot data presented in the main figures and supplementary figures as a related manuscript file entitled "Immunoblots".

4) The second paragraph of the discussion states: "... UPRRE mainly functions to attenuate protein synthesis in the ER". If protein synthesis happens in the ER, I have been misleading my students for many years!

We appreciate the humor and certainly apologize for this obvious misstatement. It has been corrected.

5) There is an obvious control not presented that hopefully has already been done: a demonstration that XBP-1s does not alter transcription of the tau transgene. It is not far-fetched that the aex-3 promoter could be influenced by the XBP-1 transcription factor. While I am listing this as a minor point, this experiment needs to be presented.

We now include quantitative RT-PCR data demonstrating no significant change in human tau transgene encoded mRNA levels (Supplementary Fig. 6).

6) It would be helpful if the resubmission included page numbers.

We now include both page numbers and line numbers to facilitate reading and editing of the manuscript.

Reviewer #3 (Reviewer's remarks are in italics; Response by the Authors are in regular type):

*While it has been convincingly demonstrated that ER stress and the unfolded protein response are activated in Alzheimer's disease and other tauopathies, the details of this relationship have remained mostly unknown. A similar relationship has also been described for other neurodegenerative diseases. Thus, this study by Waldherr et al. that seeks to evaluate if one or more of the branches of the UPR within the ER can offer protection against tau-mediated toxicity and associated phenotypes is highly relevant and the findings are potentially broadly applicable across many diseases hallmarked by protein aggregates. The authors used a genetic approach in *C. elegans* to systematically delete the regulators of this feedback system independently or in combination to better understand their connection to tau pathogenesis. They evaluated the outcomes by measuring motor behavior, neuronal counts, and Western blots. Overall, this work has revealed a very interesting and important finding that XBP1 significantly contributes to tau pathogenesis, in that in the absence of this gene tau toxicity and related phenotypes are worsened and in the presence of a constitutively active splice variant of XBP1 there is protection. The data also clearly show a role for the ATF6 and PERK branches in regulating tau pathogenesis through XBP1s. However, when silenced alone, PERK (PEK) deletion had no effects on tau-LOW phenotypes, while ablation of ATF6 worsened tau-associated phenotypes without affecting total tau levels. Overall, this is a fairly thorough study that supports the authors' conclusions. I only have a few comments that should be addressed.*

1. *Quantification should be shown for all Western blots, including those in the supplement.*

a. *Figure 3D would be more informative with a pTau/total tau ratio shown to determine any specific pTau changes*

As much as was possible in the timeframe required for this response, we now provide additional quantification of immunoblots and calculations of pTau/total tau ratios for Fig. 3d (quantification data provided in Supplementary Table 3) as requested by the Reviewer (for quantification of immunoblots presented in the main figures, see Fig. 1e, 2c, 3c, 4d and Supplementary Table 3).

2. *The authors suggest that the effects are through altered ERAD. A measurement of ERAD function should be provided to support these claims.*

We provide additional experimentation showing that XBP-1s-mediated suppression of tauopathy requires ERAD function (see Fig. 5e, and our responses to Reviewer 1 regarding the Model/Discussion-Based Comments, point "1)", and Reviewer 2 regarding Major Point "2)" addressing a similar issue above).

3. *Further discussion about the discrepancy between the XBP1s behavioral defect and the XBP1s/high tau behavioral rescue should be included.*

The *xbp-1s* Tg animals exhibit mild behavioral deficits as measured by swimming assay (~50 thrashes/min motility in liquid; see Supplementary Fig. 5), which is contrasted with the severe deficit seen in Tau (high) animals (~5 thrashes/min; see Fig. 3a); this highlights the strength of the rescue of the tauopathy behavioral phenotype by the *xbp-1s* Tg (~40 thrashes/min; see Fig. 3a), Despite *xbp-1s* Tg displaying very mild defects compared to non-Tg, the *xbp-1s* transgene rescues the very severe defect seen in Tau (high) animals. We have indicated non-Tg motility on all graphs as requested by Reviewer 1 to more clearly illustrate differences from non-Tg (see response to General Experimental Comments "1)" above, and Fig. 1b, 1c, 2a, 3a, 4a, 4b, 5e; Supplementary Fig. 3, 7a)). We also expand our discussion on this point (page 15, line 403-426).

4. *Even though it is outside the scope of the current study to investigate, the inclusion of a discussion of what genes are activated by XBP1s constitutive activation that could regulate tau turnover would be beneficial.*

We now clarify our discussion of the nature of known XBP-1s target genes and speculate about potential target genes responsible for suppression (page 15, lines 443- 466). However, this is an interesting area of ongoing inquiry we are pursuing as actively as current funding constraints permit.

References

Guthrie, C.R., Schellenberg, G.D. and Kraemer, B.C. (2009) SUT-2 potentiates tau-induced neurotoxicity in *Caenorhabditis elegans*. *Hum Mol Genet*, **18**, 1825-1838.

Kraemer, B.C., Burgess, J.K., Chen, J.H., Thomas, J.H. and Schellenberg, G.D. (2006) Molecular pathways that influence human tau-induced pathology in *Caenorhabditis elegans*. *Hum Mol Genet*, **15**, 1483-1496.

Kraemer, B.C., Zhang, B., Leverenz, J.B., Thomas, J.H., Trojanowski, J.Q. and Schellenberg, G.D. (2003) Neurodegeneration and defective neurotransmission in a *Caenorhabditis elegans* model of tauopathy. *Proc Natl Acad Sci U S A*, **100**, 9980-9985.

REVIEWERS' COMMENTS:

Reviewer #1 (Remarks to the Author):

We feel that the authors have appropriately satisfied our concerns and recommend the manuscript for publication.

Reviewer #2 (Remarks to the Author):

The manuscript has been significantly improved, and my major concerns have been addressed. I would have preferred if the investigators had also tested the effects of mutations that inhibited autophagy in their tau/xbp-1 studies, but as it stands the study is still well-supported and an important contribution to the field.

Reviewer #3 (Remarks to the Author):

The authors were very responsive to the first critiques and sufficiently addressed each of my prior concerns. I support publication of this manuscript.